Symplectin evolved from multiple duplications in bioluminescent squid

Francis Warren R. 1 2
Christianson Lynne M. 1
Haddock Steven H.D. haddock@mbari.org 1
1 Monterey Bay Aquarium Research Institute , Moss Landing , CA , United States of America
2 Department of Biology, University of Southern Denmark , Odense , Denmark
Toonen Robert
Electronic publication date: 2017 Jul 31
Publication date: 2017
Volume: 5
Electronic Location ID: e3633
Received 2017 May 22; Accepted 2017 Jul 11
Copyright: ©2017 Francis et al.
Copyright year: 2017
Copyright holder: Francis et al.
License: This is an open access article distributed under the terms of the Creative Commons Attribution License, which permits unrestricted use, distribution, reproduction and adaptation in any medium and for any purpose provided that it is properly attributed. For attribution, the original author(s), title, publication source (PeerJ) and either DOI or URL of the article must be cited.
License URL: https://creativecommons.org/licenses/by/4.0/

Keywords: Luciferase, Neofunctionalization, Coelenterazine, Squid, Gene duplication, Symplectin, Evolution, Bioluminescence

Funding: NIH NIGMS-5-R01-GM087198 This work was supported by NIH grant NIGMS-5-R01-GM087198 to S.H.D.H. The funders had no role in study design, data collection and analysis, decision to publish, or preparation of the manuscript.

==============================
The squid Sthenoteuthis oualaniensis, formerly Symplectoteuthis oualaniensis, generates light using the luciferin coelenterazine and a unique enzyme, symplectin. Genetic information is limited for bioluminescent cephalopod species, so many proteins, including symplectin, occur in public databases only as sequence isolates with few identifiable homologs. As the distribution of the symplectin/pantetheinase protein family in Metazoa remains mostly unexplored, we have sequenced the transcriptomes of four additional luminous squid, and make use of publicly available but unanalyzed data of other cephalopods, to examine the occurrence and evolution of this protein family. While the majority of spiralians have one or two copies of this protein family, four well-supported groups of proteins are found in cephalopods, one of which corresponds to symplectin. A cysteine that is critical for symplectin functioning is conserved across essentially all members of the protein family, even those unlikely to be used for bioluminescence. Conversely, active site residues involved in pantetheinase catalysis are also conserved across essentially all of these proteins, suggesting that symplectin may have multiple functions including hydrolase activity, and that the evolution of the luminous phenotype required other changes in the protein outside of the main binding pocket.

Introduction

Luminous cephalopods (squids and octopods) generate light through mechanisms both native to the animal and through control of symbiotic bacteria (Haddock, Moline & Case, 2010). Although two independent evolutionary events of bacterial bioluminescence have been identified in squid (Lindgren et al., 2012), native bioluminescence (also called autogenic bioluminescence) is by far more common. The bioluminescence mechanisms of many of these autogenic species are unknown, although it is known that most of the species make use of the same luciferin, coelenterazine, which is the consumable substrate for the reaction. Coelenterazine is the most widely occurring luciferin in marine bioluminescence, its use being report in at least eleven phyla (Haddock, Moline & Case, 2010). Some species have been found to use forms of coelenterazine with additional modifications: the firefly squid Watasenia scintillans utilizes sulfated coelenterazine (Inoue et al., 1975), while the squid Sthenoteuthis oualaniensis (Takahashi & Isobe, 1994) and the clam Pholas dactylus (Tanaka, Kuse & Nishikawa, 2009) use a version that is dehydrated at the 2-position, called dehydrocoelenterazine (hereafter abbreviated as dhCtz).

Much of the work on coelenterazine-using bioluminescent systems has focused on cnidarians (both hydromedusae like Aequorea victoria or Obelia spp. and the octocoral Renilla reniformis) and crustaceans like Gaussia princeps and Oplophorus. Nonetheless, some key protein components have been identified in two squid species. The luciferase of the firefly squid Watasenia scintillans was recently identified (Gimenez et al., 2016), belonging to the same protein superfamily as firefly luciferases. The other protein comes from the purpleback flying squid Sthenoteuthis oualaniensis, where a 500-amino acid photoprotein has been cloned and characterized (Isobe et al., 2008) and was named symplectin. While the family of firefly luciferases and marine coelenterazine luciferases has been well studied, little attention has been given to the distribution and origins of symplectins.

Previous work highlighted the sequence similarity of symplectin to members of the biotinidase/pantetheinase family (Fujii et al., 2002), part of the superfamily of carbon-nitrogen hydrolases. Such proteins have been characterized in mammals, and in Drosophila (Swango & Wolf, 2001) and have roles in recycling of the enzymatic cofactors biotin and pantothenic acid. However, information about this protein family is limited outside of model organisms. Thus, very little insight could be offered to explain how a biotinidase evolved into a photoprotein, and how this apparently happened independently of other mechanisms of bioluminescence in squid.

Transcriptomics has proven helpful in identifying new fluorescent proteins (Hunt et al., 2012) and photoproteins (Powers et al., 2013) in organisms where traditional cloning approaches have failed. Based on this, we sought to identify orthologs of symplectin in transcriptomes of other luminous squid. We present transcriptomes from light-producing tissues of four luminous squid, Chiroteuthis calyx, Octopoteuthis deletron, Vampyroteuthis infernalis, Pterygioteuthis hoylei, and one luminous nudibranch Phylliroe bucephala. We found homologs of symplectin in all five species, and then further compared them to homologs identified in other animal phyla. Many important catalytic residues are conserved across the entire protein family, including among cephalopod proteins. As this protein family has undergone many independent expansions in several other groups since the last common ancestor of animals, it may be that large expansions of protein families facilitate evolution of novel phenotypes, including bioluminescence.

Materials and Methods

Specimens and sequencing

Specimens of Chiroteuthis calyx, Octopoteuthis deletron, and Vampyroteuthis infernalis were collected around the Monterey Bay, off the coast of California using detritus samplers on ROVs (remotely-operated vehicles) from the Monterey Bay Aquarium Research Institute (MBARI). Pterygioteuthis hoylei and Phylliroe bucephala were caught in the Gulf of California by trawls and blue-water diving, respectively. ROVs were operated during daytime hours, between 07:00 and 19:00 h. Operations were conducted under permit SC-4029 issued to SHD Haddock by the California Department of Fish and Wildlife. Species used are unprotected and unregulated, and no vertebrates or octopus were used, so the International and NIH ethics guidelines are not invoked, although organisms were treated humanely. All samples were flash-frozen in liquid nitrogen. Total RNA was extracted with the Qiagen RNA-easy kit following manufacturer’s instructions. All samples were sequenced at the University of Utah using the Illumina HiSeq2000 platform. Libraries were generated using the Illumina TruSeq kit, with oligo-dT selection, and were run with six samples per lane. Assemblies were generated as previously described (Francis et al., 2013). NCBI SRA numbers are given in Table 1. All assemblies can be downloaded at https://bitbucket.org/wrf/squid-transcriptomes.

Table 1 Transcriptomic data sources.

Species	Tissue	Luminous	Bases (Gb)	Assembled transcripts	Accession	Reference	
Chiroteuthis calyx	Arm	Yes	6.3	78,445	SRR5527417	This study	
Octopoteuthis deletron	Arm tips	Yes	6.1	122,672	SRR5527415	This study	
Vampyroteuthis infernalis	Arm tips	Yes	6.4	149,961	SRR5527416	This study	
Pterygioteuthis hoylei	Photophore	Yes	6.8	93,201	SRR5527418	This study	
Dosidicus gigas	Photophore	Yes	6.0	94,197	SRR5152122	Francis et al. (2013)	
Watasenia scintillans	Arm/mantle	Yes	30.0	216,307	GEDZ00000000.1	Gimenez et al. (2016)	
Uroteuthis edulis	Photophore	Bacterial	8.8	119,033	PRJNA257113	Pankey et al. (2014)	
Euprymna scolopes	Multiple	Bacterial	45.6	280,433	PRJNA257113	Pankey et al. (2014)	
Octopus vulgaris	Nerve/brain	No	1.2	59,859	JR435555	Zhang et al. (2012)	
Sepia pharaonis	Whole animal	No	4.6	131,176	SRR3011300	Wen et al. (2016)	
Loligo vulgaris	Sucker	No	3.1	43,951	SRR3472303	Jung et al. (2016)	
Doryteuthis pealeii	Multiple	No	15.2	212,516	PRJNA255916	Alon et al. (2015)	
Phylliroe bucephala	Whole animal	Yes	6.1	66,535	SRR5527414	This study	

Data acquisition

We made use of public data from a number of previous studies (Table 1). When available, assembled transcriptomes were used. Otherwise, data were downloaded from NCBI SRA and assembled with Trinity v2.2.0 (Grabherr et al., 2011), with the options –normalize_reads and –trimmomatic.

Gene trees

Homologs of symplectin were identified by BLAST alignment using blastp or tblastn, using symplectin (NCBI accession: C6KYS2.2) as the query and an e-value threshold of 10−10. All BLAST searches were done using the NCBI BLAST 2.2.29 + package (Camacho et al., 2009). As the query was 501 amino acids, partial proteins under 100 amino acids were unlikely to provide useful comparisons and were excluded. Alignments for protein sequences were created using MAFFT v7.029b, with L-INS-i parameters for accurate alignments (Katoh & Standley, 2013). Phylogenetic trees were generated using either FastTree (Price, Dehal & Arkin, 2010) with default parameters or RAxML-HPC-PTHREADS v8.2.10 (Stamatakis, 2014), using the PROTGAMMALG model for proteins and 100 bootstrap replicates with the “rapid bootstrap” (-f a) algorithm.

Structure modelling

A model structure of symplectin was generated using the HHPred webserver (Alva et al., 2016), with human vanin-1 (PDB accession: 4CYG) as the template protein, using default parameters. Vanin-1 was the highest scoring model available (probability of 100, e-value of 1.4∗10−71). Model evaluations can be downloaded at https://bitbucket.org/wrf/squid-transcriptomes.

Results

Symplectin-like proteins in other cephalopods

We generated approximately 6 Gb of pair-end RNAseq data for 4 squid and one luminous nudibranch (Table 1). Reads for each species were assembled de novo using Trinity (Grabherr et al., 2011). With symplectin as the query, we used BLAST (Camacho et al., 2009) to identify homologs in the transcriptomes and genomes of other animals. In general, cephalopods have four copies of this protein sharing a single origin (Fig. 1), while the precise number varies between species, likely subject to coverage limits of transcriptome sequencing or tissue-specific expression. We found full-length proteins of each of the four groups from only three species, Watasenia scintillans, Sepia pharaonis (non-luminous), and Pterygioteuthis hoylei. The symplectin group and group 1 proteins were not found in any octopodiform (represented by only V. infernalis and two Octopus species) (Zhang et al., 2012; Albertin et al., 2015).

Figure 1 Phylogenetic tree of symplectin and homologs from other molluscs.

Four strongly supported groups are identified in cephalopods. Species that are luminous but have unknown mechanisms are high-lighted in blue, while those with bacterial bioluminescence or other known proteins are highlighted in orange and pink, respectively. Black labels indicate non-luminous species. Clades/proteins with a catalytic triad of E-K-S have a green star, and those with E-R-S triad have a yellow star; all others have the conserved E-K-C triad. Internal bootstrap values are removed for clarity. Complete version of the same tree containing homologs from across metazoans is shown in Fig. 2.

Figure 2 Phylogenetic tree of known symplectin and homologs from all animals.

Cephalopod-specific groups are indicated as in Fig. 1. Symplectin is indicated by the blue star. Most species have only a single homolog and the proteins form a clade by phylum. This is not true of chordates (yellow and orange for biotinidase and vanin-1/pantetheinase, respectively) and cnidarians (red). Internal bootstrap values are removed for clarity, though many support values for backbone nodes are low. Detailed version of the same tree showing only molluscs is shown in Fig. 1.

Of the four protein groups, homologs of symplectin were found in two luminous species where the mechanisms of light production are unknown, D. gigas and P. hoylei. These homologs could potentially be the active protein in the bioluminescence of these species. However, the firefly squid W. scintillans has five proteins in total, one of each of the four groups, as well as an additional duplicate of the symplectin group. We were also able to find symplectin homologs in three species that are not luminous, Sepia pharaonis, Loligo vulgaris, and Doryteuthis pealei, indicating that the tree position alone is not enough to predict bioluminescence of the organism, or whether the enzyme could act as a photoprotein; it could be that all four cephalopod protein groups can act as photoproteins, though it could also be that symplectin is the only enzyme that acts as a photoprotein in this entire protein family. Additionally, two species that have luminous systems which do not involve coelenterazine still have members of this protein family: the squid Euprymna scolopes and Uroteuthis edulis both generate light from interactions with symbiotic, luminous bacteria, yet we still found homologs from protein groups 1, 2 and 3.

Symplectin-like proteins across metazoans

To better understand the relative importance of the cephalopod duplications, we examined homologs of symplectin across Metazoa (Fig. 2). At the sequence level, symplectin is similar to two annotated proteins in human (Fujii et al., 2002), vanin-1 (pantetheinase, 30% identity) and biotinidase (btd1, 31% identity), both of which are hydrolyzing amidases and do not require other cofactors. Homologs of symplectin/biotinidase were found in most animal groups, including vertebrates, arthropods, cnidarians, sponges, placozoans, as well as choanoflagellates (single-celled eukaryotes that are sister-group to animals). We were unable to find any homologs in hemichordates (based on the genomes of Ptychodera flava and Saccoglossus kowalevskii) nor any ctenophores (based on the genome of Mnemiopsis leidyi and transcriptomes from 11 species). Because of the presence in choanoflagellates, absence of this protein family in hemichordates and ctenophores is likely a secondary loss. However, hemichordates are only represented by two species so sequencing of additional species, or deeper sequencing of the studied species, may reveal homologs in this clade.

Comparison to annotated proteins

It is clear from the alignment that the original symplectin sequence is not the complete CDS (Fig. 3), as it does not start with methionine, and around 30 residues are missing compared to symplectin homologs in other cephalopods. At present, no transcriptomic or genomic data is available for S. oualaniensis, thus we could not examine the protein completeness or copy number in that species.

Figure 3 Alignment of symplectin and vanin-1.

Multiple sequence alignment of symplectin and vanin-1 with top hits from D. gigas, P. hoylei, and W. scintillans. Intensity of blue color shows conservation. Catalytic residues (E, K, C) identified in vanin-1 are indicated by triangles beneath, and the catalytic cysteine is shown in green, though this position is a serine for D. gigas. The dhCtz-binding cysteine is shown in yellow, indicated by the blue hexagon above. Disulfide bridges found in both symplectin and vanin-1 are shown in black. Those found only in vanin-1 are shown in green, while the one remaining disulfide bond found in symplectin is shown in red. For Group 2 cephalopod proteins (Fig. 1), the conserved cysteine at alignment position 440 is substituted and a neighboring residue (I383 in symplectin) is instead a cysteine.

While symplectin does not have an available crystal structure, the structure of the human protein vanin-1 has been determined (Boersma et al., 2014), which is the only member of this family to have a crystal structure. The vanin-1 protein structure is divided into two domains, the catalytic “nitrilase” domain, and the base domain, which has an unknown function in vanin-1 (Boersma et al., 2014). The catalytic triad residues of human vanin-1 include a glutamate (E79), lysine (K178) and cysteine (C211) (Boersma et al., 2014). Two of the residues, K178 and E79, are conserved in essentially all taxa, including cephalopods (though this cannot be evaluated in partial sequences). The cysteine is also conserved in most proteins, except some cephalopods sequences have a serine instead of cysteine, although serine could still function as a nucleophile for hydrolysis. All Group 3 proteins are serine-containing, but otherwise the serine proteins do not form a clade, indicating that this mutation has occurred multiple times independently. The cysteine is followed by a phenylalanine (F212 in vanin-1) in nearly all proteins, while most of the cephalopod serine proteins are followed by tyrosine, suggesting some role of this position in the specificity or reactivity of the enzyme.

Conservation and role of cysteines

Instead of using the more-common coelenterazine molecule, symplectin requires dehydrocoelenterazine (dhCtz). Unlike cnidarian or ctenophore photoproteins, which hold coelenterazine through a peroxide bond to tyrosine (Head et al., 2000), dhCtz is linked to symplectin by a thioether to a free cysteine residue (C390 in Symplectin) (Isobe et al., 2008). Symplectin has 11 cysteines (Fig. 3), initially suggesting that the odd number allowed for a free cysteine to bind coelenterazine. However, mass spectrometry data indicate that only three pairs are in the form of disulfide bridges (Kongjinda et al., 2011), though perhaps the remaining two pairs are not effectively captured. If the conserved cysteine (C196) is actually catalytic in the binding pocket of the nitrilase domain of the symplectin and does not form a bridge, and three pairs are already identified, then this leaves one additional pair of cysteines for disulfide bonding.

C390 is conserved in essentially all other proteins of this family, including biotinidase (C471) and vanin-1 (C411). In vanin-1, C411 is in close proximity to C403 and forms a disulfide bond, though apparently this cannot occur in symplectin between C385 and C390, since C390 would no longer be available for thioether bonding to dhCtz (Isobe et al., 2008). Mass spectrometry data suggest that C385 instead forms a bridge with C380 (Kongjinda et al., 2011). However, when symplectin is modeled based on the structure of vanin-1 (Fig. 4), C380 would be on another beta strand and point away from the binding pocket to bridge with C345, while the adjacent cysteine (C344) bridges further away to C339 in a beta-hairpin motif. The cysteine pair for C345/C380 is found in all proteins suggesting an important structural role of C380, and arguing against the formation of the C380/C385 bond. By comparison, the C344/C339 pair is not found in sponges, polychaetes, or choanoflagellates, suggesting it is dispensable.

Figure 4 Modeled structure of symplectin.

PDB format structure based on the structure of vanin-1 (4CYF). (A) Overview of the two domains, named after those defined in vanin-1. Residues 1–290 (blue to green) compose the nitrilase domain while residues 291–465 compose the base domain (yellow to red). (B) Close-up view of the catalytic triad E60-K163-C196 (C) Putative disulfide bridge of C390 and C385, while overall these residues are located close to a number of hydrophobic residues, potentially involved in dehydrocoelenterzine binding.

This discrepancy may be reconciled three ways: (1) the structures of symplectin and vanin-1 differ enough that vanin-1 cannot reliably be used to model disulfide bridges in symplectin; (2) C385/C390 natively form a disulfide bond, though when digested, other changes in the redox state disrupt this bond and the portion containing C380 and C385 dynamically forms a disulfide bond when fragmented. However, this presents an additional problem, as even if C385 and C390 do indeed form a disulfide bridge, this bond must break in order for the thioether to form, and C385 is then a free cysteine; (3) the fragmentation data are incorrect and do not reliably capture the disulfide bridges of the native protein, and the free cysteine in the catalytic core of the nitrilase domain (C196) is actually responsible for the photoprotein activity.

Discussion

Catalytic structure predictions

Assuming the overall structure is similar between vanin-1 and symplectin, the conserved catalytic triad is located in the nitrilase domain of symplectin (E60, K163, C196), while the cysteine for the thioether linkage is found in the base domain. Therefore, catalytic binding pocket of nitrilase domain is not responsible for the bioluminescence activity of symplectin, which is instead likely carried out by residues in close spatial proximity to C390. Although two other luciferases have solved crystal structures (Loening, Fenn & Gambhir, 2007; Tomabechi et al., 2016), these were unbound forms so mechanistic generalizations cannot be made.

Nonetheless, three phenylalanines (F316, F321, and F323) and one tyrosine (Y359) are predicted to be close to C390, and may have a role in coordinating the binding of coelenterazine, although only the tyrosine is well-conserved outside of symplectin. Besides the non-polar residues, several other residues predicted to be in proximity to C390 may have a role in the catalytic activity, including D314, K325, and E357. All three are conserved in one protein each from P. hoylei and W. scintillans, which are the two closest proteins to symplectin in the tree. However, given that the overall conservation is low, even proteins evolutionarily close to symplectin may not have any luminescence activity.

It was also noted that the native vanin-1 structure is a homodimer (Boersma et al., 2014). Head-to-tail arrangement of the two domains within a homodimer would allow another alternative, where the nitrilase domain of one monomer catalyzes the oxidation of dhCtz bound to the other monomer. In this case, there may be no need for any residues directly surrounding C390 to have a role in the enzymatic activity.

Evolution of the protein families

Interpreting the phylogenetic tree and determining the ancestral number of genes is challenging given the copy number in some animal lineages relative to the tree position. For most cases, such as the majority of arthropods, lineage-specific duplications have created multiple copies, the most being eight homologs in the spider Parasteatoda tepidariorum. The same is seen for the two polychaetes and other non-cephalopod molluscs. However, cnidarians (only anemones and corals represented) and chordates have two separate protein groups. For chordates, pantetheinase and biotinidase each have their own group, and although the copy number relative to the cephalochordate outgroup Branchiostoma floridae suggests a duplication specific to vertebrates, the tree does not indicate this arrangement with strong support. For cnidarians, both groups have unknown functions and are not monophyletic. Clearly a duplication must have occurred to give rise to the two groups, although the copy number in other groups (such as Porifera or Placozoa) is inconsistent with a duplication in the common ancestor of all animals. Two scenarios may be considered. The two cnidarian groups resulted from a duplication specific to cnidarians even though the tree (Fig. 2) does not indicate a single origin, suggesting that this protein family is poorly resolved by existing phylogenetics programs. The same would therefore also be true for the two vertebrate proteins. Alternatively, multiple bilaterian phyla (molluscs, annelids, arthropods, echinoderms) must have lost at least one of the ancient copies that was otherwise retained by chordates.

Given the ubitquity of both biotin and pantothenic acid, metabolism of one or both of these molecules is likely to be the ancestral function. Biotinidase activity was detected in Drosophila melanogaster (Swango & Wolf, 2001), providing evidence that the monophyletic group of arthropod proteins are biotinidases, and possibly all bilaterian members of this family. This would therefore suggest several points about the evolution of this protein family. First, one or many of the cephalopod proteins may still have biotinidase activity or act as a hydrolase in other contexts via the nitrilase domain. Since symplectin is therefore predicted to have two separate functional domains, it is possible that symplectin performs multiple functions, and may even still have a role in biotin metabolism. Secondly, if the ancestral function is a biotinidase, then the pantetheinase activity of vanin-1 and related proteins found only in vertebrates therefore is a derived function.

Among the four cephalopod protein groups, homologs of the symplectin group were still found in several non-luminous species, Sepia pharaonis, Loligo vulgaris, and Doryteuthis pealei. For this reason, it could be that only a very small set of symplectin homologs are luminous because of key amino acid changes in the protein. While we were not able to find all four homologs in the transcriptomes of most species, we found five in W. scintillans, where two were in the symplectin group. Of these, one was highly similar to symplectin, including a symplectin-specific indel of two amino acids found in only one other protein (in P. hoylei). Candidate luciferases were already identified from W. scintillans (Gimenez et al., 2016) belonging to another protein family, so it is unclear what the role would be of the symplectin homologs identified here, or whether they have photoprotein activity with dhCtz.

Bioluminescence in cephalopods

Phylogenetic analysis of cephalopods suggests a complex pattern of gains and losses of bioluminescence (Lindgren et al., 2012), where a luminous phenotype appears to have been acquired five times, and pelagic species were significantly more likely to display autogenic bioluminescence. In addition to the two independent gains of bacterial bioluminescence, there are a total of seven inventions of bioluminescence in this class. As the protein mechanisms are only known for two species, W. scintillans and S. oualaniensis, is it possible that several other luciferases have evolved in this family.

Convergent evolution of bioluminescence is comparably commonplace within other animal groups. For instance, cnidarians are likely to have at least five separate evolutionary events of bioluminescence, in octocorals, deep-sea anemones (Johnsen et al., 2012), coronate and semaeostome scyphomedusae (Shimomura et al., 2001) and hydromedusae, though the proteins responsible have only been identified in octocorals (Renilla-type luciferases) and hydromedusae (calcium-activated photoproteins). Thus, even within the same clade, multiple separate origins of bioluminescence making use of the same luciferin with different proteins has already happened at least once in metazoans.

For the proteins themselves, three inventions of luciferases have been found in family of adenylating enzymes: for fireflies, the New Zealand glow worm Arachnocampa luminosa (Sharpe et al., 2015), and the squid W. scintillans (Gimenez et al., 2016). This may suggest that members of this protein family has some propensity for becoming luciferases. Because many luciferins are hydrophobic molecules, enzymes that already catalyze reactions on other hydrophobic molecules (such as fatty acids) may have some innate affinity for luciferins, and then mutations that allow binding of oxygen ultimately change the function of these enzymes into luciferases.

Future directions

The case of convergent evolution of luciferases from adenylating enzymes was essentially unpredictable. For this reason, it is just as plausible to consider that none of the symplectin homologs here can function as luciferases as it is to consider that all of them function this way. Cloning and characterizing all symplectin-group homologs in vitro may resolve this, but does not effectively account for the possibility where luciferase activity independently evolved twice from the same protein family. Thus, if it were necessary to clone all members of a protein family to check for luciferase activity, this may be prohibitively time consuming. For this reason, a more directed approach of proteomic investigations of bioluminescent material from certain species may prove to be more useful, particularly for a species like the vampire squid, which has a secreted bioluminescence (Robison et al., 2003) and would be comparably easy to acquire concentrated luminous material. Such studies typically require a reference genome or transcriptome, so potentially the transcriptomes presented here could serve as a reference for these or closely related species. Further investigation of the mechanisms of bioluminescence in this animal clade may reveal more general principles of protein evolution, and how a few amino acid changes may have dramatic effects on the phenotype and biology of an organism.

Supplemental Information

Supplemental Information 1 RAxML output tree in Newick format with bootstrap values

Click here for additional data file.

Supplemental Information 2 Multiple sequence alignment of symplectin and homologs

Click here for additional data file.

Supplemental Information 3 Unaligned proteins for alignment

Click here for additional data file.

Additional Information and Declarations

Competing Interests

Author Contributions

Field Study Permissions

DNA Deposition

Data Availability

The authors declare there are no competing interests.

Warren R. Francis conceived and designed the experiments, performed the experiments, analyzed the data, contributed reagents/materials/analysis tools, wrote the paper, prepared figures and/or tables.

Lynne M. Christianson performed the experiments.

Steven H.D. Haddock conceived and designed the experiments, analyzed the data, wrote the paper, prepared figures and/or tables.

The following information was supplied relating to field study approvals (i.e., approving body and any reference numbers):

Operations were conducted under permit SC-4029 issued to SHD Haddock by the California Department of Fish and Wildlife.

The following information was supplied regarding the deposition of DNA sequences:

Reads for all five transcriptomes are at NCBI SRA with accessions SRR5527414 –SRR5527418.

The following information was supplied regarding data availability:

All assemblies, alignments, and intermediate files are deposited at BitBucket: https://bitbucket.org/wrf/squid-transcriptomes.

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
