# Peer review of "Symplectin evolved from multiple duplications in bioluminescent squid"

_PeerJ, doi:10.7717/peerj.3633_

## Round 0.1 · original submission · Minor Revisions

I am sorry for the delay in getting this manuscript back to you, we have been waiting on a final review for your feedback, but rather than delay this process for you any longer, we have decided to proceed with just the 2 reviews in hand. Both are equally supportive of the work, and each have only relatively minor suggestions for improvement of the manuscript overall that I do not foresee any difficulty in addressing. As such, I expect that you should be able to turn this around relatively easily with the suggested clarifications and corrections taken into consideration, and I look forward to receiving your revised manuscript.

·

Basic reporting

No comment.

Experimental design

Details on BLAST parameters missing. Need to report e-value thresholds or minimum sequence similarities.
Some details relevant to Figure 2 are missing or unclear: this tree was generated by BLASTing genomes and transcriptomes? Was the same BLAST threshold used for Figures 1 and 2?

Validity of the findings

No comment.

Additional comments

Minor revisions:
l.42 More background in intro: What roles do pantotheinase and biotinidases play in the well-characterized models?
l 60. Methods: how were samples preserved (-80, RNAlater etc)?
l.103: Avoid speculation in results

Table and l. 112: Uroteuthis 'enopla' should be Uroteuthis edulis.

l.97-113: Big caveat to testing hypotheses of molecular evolution with transcriptomes is that failure to detect a paralog's transcript does not imply its genomic absence. Must be careful to
avoid this interpretation in stating results: For instance,some species are represented by multiple tissues..this could artifactualy

increase the chance of detecting symplectin paralogs.

l.127-148: Avoid speculation in results

l.164 define 'dhCtz'

Figure 1. colors confusing: purple has two meanings: symplectin paralog and 'other proteins' mechanism?

Fig. 2: Is this meant to demonstrate that that symplectin is a member of the vanin family (and thus justify using the vanin structure for modelling), with pantotheinase as the outgroup?
There are no support values to indicate this hypothesis is supported.

The color scheme is confusing. Color schemes mis-match between Fig 1 and 2. Either make symplectin group same color in both for
clarity, or use entirely different palettes for each figure. Might be easier to follow authors' interpretation if rooted tree is rooted at choanoflagellates. In the legend, remind reader that btd=biotinidase and vanin=pantotheinase

Reviewer 2 ·

Basic reporting

Manuscript meets editorial criteria for basic reporting with the following
suggestion for improvement:

Introduction
1) Line 42. Do pantetheinases and biotinidases belong to the same superfamily, and if so, what is the name of this superfamily and its general characteristics? This is useful introductory information from a symplectin structural point of view.

Experimental design

Manuscript meets editorial criteria for experimental design with the following suggestions for improvement:

Materials and methods
1) Line 80. Include the accession number for the symplectin sequence used.
2) Lines 88-89. Include the PDB accession ID for the vanin-1 structure used in modelling here at first mention (even though it is mentioned elsewhere).
Results
3) Line 94. What was the cut-off used to define homologs in the BLAST search?

Validity of the findings

Manuscript meets editorial criteria for validity of the findings with the following suggestions for improvement:

Results
1) Line 94. What was the cut-off used to define homologs in the BLAST search?
2) Line 116. “In comparisons...” please re-phrase this sentence to make its meaning clearer.
3) Line 117. State the % amino acid sequence identities of the known sequence of symplectin with these two proteins.
4) Consider moving the first reference to Figure 2 from line 139 to earlier in the subsection “Symplectin-like proteins across metazoans”, to help readers better interpret the associated results.
5) Lines 122-4 “Because ... loss”. This statement is possibly a little strong for the evidence available. Consider adding “likely” or “probably” to “due to secondary loss”.
6) Line 178. Please comment on how trustworthy this model is, eg using the % identity between symplectin and vanin-1 and perhaps any measure of confidence the HHPred program provides.
7) Figure 4. The structure in Fig 4A is nicely pictured, but please label residues mentioned in the text as well as C390 in this image, including the catalytic triad of the nitrilase domain, to enable the reader to picture where these residues are positioned to each other. The residues in Figs 4B and C are hard to see. Please make them different colours to the backbone cartoon representation, and label them.

Discussion
8) Line 196. It may be more accurate to name this subsection “Catalytic structure predictions”, or something similar, since the function of the protein is already known (catalysis of bioluminescence), and the subsection speculates on which residues may be involved in catalytic activity.
9) Lines 201-202 “Although two other luciferases have solved crystal structures...” This sentence is inaccurate and needs adjustment and elaboration. The authors possibly mean that the structures of two other luciferases that use coelenterazine have been solved. Actually, there have now been three different types of coelenterazine-utilizing luciferase structures solved, the Oplophorus and the Renilla luciferases referenced here, and also the hydrozoan photoproteins (aequorin, obelin and clytin). For more details on the Renilla and hydrozoan types of structures, see the review: Sharpe, Hastings, and Krause (2014) Luciferases and Light-emitting Accessory Proteins: Structural Biology. In: eLS. John Wiley & Sons, Ltd: Chichester.
DOI: 0.1002/9780470015902.a0003064.pub2
A photoprotein is actually an intermediate-bound luciferase enzyme that is accumulated in the absence of a final reactant (calcium), and is then discharged rapidly when calcium is added (producing a flash of light), so should be included in luciferase discussions.
“...these were unbound forms so mechanistic generalizations cannot be made” Actually, it has been shown that aequorin uses the triad of tyrosine, tryptophan and histidine for catalysis, and the three main catalytic residues in Renilla luciferase are thought to be aspartic acid, histidine and glutamate. Speculation on the catalytic residues of symplectin should take this information into account.
10) Line 203. Consider labelling these residues in Fig 4A.
11) Line 228. Please elaborate on what “derived” means in this context.
12) Line 266. Please provide more detail on what these “proteomic investigations” would entail, so the reader can judge if they will be useful.

Additional comments

This thoughtfully written manuscript describes the search for homologous sequences of the symplectin squid luciferase in transcriptomes from other cephalopod and non-cephalopod species. It also carries out a cursory structural analysis of the symplectin protein using a homology-based three-dimensional structural model.
The analyses of symplectin and homologs are a useful contribution to knowledge of evolution of both bioluminescence and of enzymes generally. The sequencing and assembly work described within will be of interest for other genetic studies of cephalopods. The structural analysis of symplectin raises some interesting possibilities with regard to how the enzyme might function, which I look forward to being resolved by future studies. A little more attention to the figure featuring the structural homology model and to the discussion will result in a paper ready for publishing.

Other suggestions for improvement:
Abstract
1) Please consider adding the common name of the squid Sthenoteuthis oualaniensis (purpleback flying squid?) to help enable more potential readers to find the paper during literature searches.

Typographical errors:
2) Line 12. Do you mean “four distinct groups of these proteins”? Please clarify.
3) Line 15. Insert “catalysis” after “pantetheinase”, and “of these” after “all”.
4) Line 42. Check spelling of pantetheinase/pantotheinase throughout manuscript.
5) Line 69. Insert “the” before “Qiagen”.
6) Line 189. Change “addition” to “additional”.
7) Line 192. Add “the” before “nitrilase”.
8) Line 211. Remove “a” from in front of “another”.

---

## Round 0.2 · accepted · Accept

Having read through the original reviews from the referees and
your responses and your responses, I am satisfied with your revisions, and am happy to move your manuscript into production. Thank you for your care in responding to the referee comments.